Re-evaluation of mastodon material from Oregon and Washington, USA, Alberta, Canada, and Hidalgo and Jalisco, Mexico

Dooley Alton C. Jr 1
http://orcid.org/0000-0002-2961-0078 Widga Chris 2
Stoneburg Brittney E. 1 bstoneburg@westerncentermuseum.org
http://orcid.org/0000-0001-6499-5260 Jass Christopher 3
Bravo-Cuevas Victor M. 4
Boehm Andrew 5
http://orcid.org/0000-0002-2730-0893 Scott Eric 6
http://orcid.org/0000-0002-0324-0697 McDonald Andrew T. 1
Volmut Mark 7
1 Western Science Center , Hemet, California , United States
2 College of Earth and Mineral Sciences, Pennsylvania State University , University Park, Pennsylvania , United States
3 Royal Alberta Museum , Edmonton, Alberta , Canada
4 Museo de Paleontología, Universidad Autónoma Hidalgo , Pachuca, Hidalgo , Mexico
5 Museum of Natural and Cultural History, University of Oregon , Eugene, Oregon , United States
6 Cogstone Resource Management , Riverside, California , United States
7 Faunal Archaeology Consultant , Olympia, Washington , United States
Heritage Steven
Electronic publication date: 2025 Jan 23
Publication date: 2025
Volume: 13
Electronic Location ID: e18848
Received 2023 Dec 29; Accepted 2024 Dec 19
Copyright: © 2025 Dooley Jr et al.
Copyright year: 2025
Copyright holder: Dooley Jr et al.
License: This is an open access article distributed under the terms of the Creative Commons Attribution License, which permits unrestricted use, distribution, reproduction and adaptation in any medium and for any purpose provided that it is properly attributed. For attribution, the original author(s), title, publication source (PeerJ) and either DOI or URL of the article must be cited.
License URL: https://creativecommons.org/licenses/by/4.0/

Keywords: Proboscidea, Mastodon, Pleistocene, Mammutidae, Washington, Oregon, Alberta, Hidalgo, Jalisco

Funding: The authors received no funding for this work.

==============================
The presence of at least two contemporaneous Pleistocene mastodon taxa in North America (Mammut americanum and M. pacificus) invites re-examination of specimens at the geographic margins of each species in order to determine range boundaries, overlaps, and fluctuations. Third molars from Oregon in the United States, as well as from Hidalgo and Jalisco in Mexico, were found to be morphologically consistent with M. pacificus. Washington in the United States includes a number of specimens that could not be confidently assigned to either taxon. Alberta in Canada was found to have some specimens that were consistent with M. pacificus, but others that were identified as M. americanum. The Alberta specimen referred to M. pacificus is the same tooth found to have a Pliocene divergence time from M. americanum based on mitochondrial genome data from a previous study, suggesting a deep divergence time between the two taxa. The apparent presence of both mastodon taxa in close geographic proximity has interesting paleobiogeographic implications. It is not yet clear if both taxa were present simultaneously in a given location; if not, it suggests fluctuating ranges that may reflect shifting climates and/or biomes over time. Alternatively, if both taxa were simultaneously present in the same place, it may suggest a high degree of niche partitioning in mammutids. Additional accurately dated specimens will be required to resolve this question.

Introduction

Mastodons (Mammut) are a nearly ubiquitous part of the Late Pleistocene fauna of North America and have been scientifically studied for more than 200 years. With such a lengthy period of study, it is perhaps surprising that recent research has revealed new information about mastodons, including unexpected regional concentrations of specimens (e.g., Springer et al., 2009, 2010; Fisher et al., 2014), information about life histories and extinctions (Fisher, 2008, 2009; Miller et al., 2022; Smith & Fisher, 2011; Widga et al., 2017, 2021), genetic information documenting complex biogeographic patterns (Karpinski et al., 2020), and previously unrecognized taxa (Dooley et al., 2019).

Dooley et al. (2019) described M. pacificus based primarily on statistically narrower M3/m3 for a given length, as well as several additional characters including (relative to M. americanum) smaller diameter male tusks for a given Laws Group age and accompanying differences in the maxillae, femora with a greater mid shaft diameter for a given length, six sacral vertebrae (4–6, typically 5 in M. americanum), and the absence of mandibular tusks (present in 27% of M. americanum).

When describing M. pacificus, Dooley et al. (2019) took a conservative approach in referring specimens to M. pacificus. They only referred specimens for which there was compelling morphological and biogeographic data to M. pacificus, while all other specimens were considered M. americanum (i.e., essentially, the null hypothesis was that a specimen was M. americanum). As a result, they considered material from the Pacific Northwest and from Mexico to be M. americanum, as at the time there was little biogeographic or morphological support for referral of these specimens to M. pacificus. Additional studies on mammutids since 2019, including McDonald et al. (2020) and Karpinski et al. (2020) invite reassessment of material from these regions.

The discovery of the Pacific mastodon (Mammut pacificus) on the west coast of North America (Dooley et al., 2019) and the high level of endemism indicated in genetic data (Karpinski et al., 2020) both suggest that much remains to be learned about the diversity and relationships of different regional populations of mastodons. Indeed, the discovery of a Pacific mastodon specimen in Montana (McDonald et al., 2020), hundreds of kilometers east of any other records of this taxon, confirmed the potential for valuable data to be derived from locations not traditionally considered as “mastodon country”.

Here we report several newly recognized occurrences of Mammut pacificus in Canada, the United States, and Mexico, based on specimens previously referred to M. americanum. Some of these represent considerable range extensions for M. pacificus, and may indicate a more complex biogeographic history of Pleistocene mastodons.

Materials and Methods

Direct measurements were taken of Mammut molars and femora of specimens from Alberta in Canada, Oregon and Washington in the United States, and Hidalgo and Jalisco in Mexico. These were compared with specimens, casts, and printed replicas of Mammut elements housed at the Western Science Center, as well as to published data on other Mammut specimens, based on taxon assignments from Dooley et al. (2019). Measurements and calculations for length/width (L:W) ratios of mastodon third molars follow Dooley et al. (2019); measurements of femora follow Hodgson et al. (2008). Methods for assessing the width of the incomplete molar from Alberta (RAM P97.7.1) are described with that specimen.

We assign a specimen to M. pacificus if it meets one of the following conditions: (a) the M3 or m3 falls outside of 2σ (σ = standard deviations) for M. americanum as described in Dooley et al., 2019, or (b) if the M3 or m3 falls outside of 1σ for M. americanum, and the specimens shares an additional character associated with M. pacificus as described in the diagnosis presented in Dooley et al. (2019), with no characters unique to M. americanum. Particularly relevant characters in this study include the femoral length to midshaft width ratio, and the presence or absence of mandibular tusks (the presence of mandibular tusks is a positive indicator for M. americanum).

Results

Oregon

Dooley et al. (2019) considered Pleistocene mammutid material from Oregon in their study as Mammut americanum. All of the teeth they examined were either non-diagnostic for distinguishing between M. americanum and M. pacificus (e.g., M2), or came from tooth positions with very small sample sizes (e.g., premolars). Even so, Oregon specimens of tooth positions with small sample sizes, such as P3, still had L:W ratios that were more similar to M. pacificus than to M. americanum. Dooley et al. (2019) noted that these remains were biogeographically and anatomically anomalous if assigned to M. americanum and hypothesized that they may in fact represent M. pacificus.

One specimen not examined by Dooley et al. (2019) was the Tualatin mastodon (F-30282), recovered in 1962 and currently on display in the Tualatin Public Library. Nearly the entire left side of the animal is preserved, including the left tusk and a portion of the maxilla with the preserved M2 and M3 (Fig. 1A). Based on photos of the excavation and rudimentary notes, the remains were situated approximately 3.5–5 feet (1.067–1.52 m) below the surface in a marsh. The Tualatin mastodon dates to the Late Pleistocene (Gilmour et al., 2015), post-dating the Missoula floods and human colonization of the area (Davis et al., 2019; O’Connor et al., 2020). The M3 has a L:W ratio of 1.91, which is just inside the range of M3s of M. americanum in our dataset (1.59–1.95), but well within the range of M. pacificus (1.69–2.33) (Table 1, Fig. 2). Additionally, the left femur (Fig. 1B) is complete and has a maximum length of 807 mm and mid-shaft width of 130 mm, placing it close to small M. pacificus specimens from California, (Dooley et al., 2019: Figure 25; Fig. 3), and well apart from M. americanum. These measurements indicate that the Tualatin mastodon is M. pacificus, suggesting that other Oregon material previously reported as M. americanum may be M. pacificus as well.

Figure 1 Mammut pacificus F-30282 (Tualatin mastodon).

(A) Left M3, occlusal view, 5 cm scale bar. (B) Left femur in articulation with pelvis, 10 cm scale bar.

Table 1 Aggregate M3 data.

State/Province	n	Mean maximum length	Median maximum length	SD	Max	Min	Mean maximum width	Median maximum width	SD	Max	Min	Mean L/W	Median L/W	SD	Max	Min		
California	39	168.76	168.00	14.79	202.77	142.50	85.39	85.20	6.00	104.26	73.08	1.98	1.96	0.14	2.33	1.69		
Montana	1	174.70	174.70		174.70	174.70	78.80	78.80		104.26	78.80	2.22	2.22		2.22	2.22		
Oregon	1	149.89	149.89		149.89	149.89	78.61	78.61		104.26	78.61	1.91	1.91		1.91	1.91		
Hidalgo	1	158.00	158.00		158.00	158.00	82.00	82.00		104.26	82.00	1.93	1.93		1.93	1.93		
Jalisco	1	168.97	168.97		168.97	168.97	81.60	81.60		104.26	81.60	2.07	2.07		2.07	2.07		
Alaska	2	157.00	157.00	10.32	164.30	149.7	95.58	95.58	2.02	104.26	94.2	1.64	1.64	0.07	1.69	1.59		
Arizona	1	188.2	188.20		188.20	188.2	98.2	98.20		104.26	98.2	1.92	1.92		1.92	1.92		
Colorado	4	163.50	163.50	0.58	164.00	163.00	99.30	99.25	1.70	104.26	97.50	1.65	1.65	0.03	1.67	1.61		
Florida	15	177.35	177.30	15.20	197.40	143.20	99.07	100.60	7.51	104.26	82.20	1.79	1.79	0.11	1.95	1.59		
Georgia	1	184.00	184.00		184.00	184.00	104.00	104.00		104.26	104.00	1.77	1.77		1.77	1.77		
Illinois	3	181.81	175.94	11.00	194.50	175.00	104.15	102.00	6.42	104.26	99.07	1.75	1.75	0.03	1.78	1.72		
Indiana	8	175.32	180.50	20.10	203.00	150.04	99.86	99.75	7.16	104.26	87.85	1.75	1.71	0.14	1.95	1.60		
Kentucky	2	164.73	164.73	5.58	168.67	160.78	91.89	91.89	7.16	104.26	86.83	1.80	1.80	0.08	1.85	1.74		
Louisiana	4	176.02	174.00	17.78	196.60	159.46	106.18	104.36	9.73	104.26	98.00	1.66	1.66	0.02	1.68	1.63		
Missouri	23	181.99	181.70	18.07	213.51	144.30	101.02	101.70	7.92	104.26	86.60	1.80	1.80	0.08	1.93	1.63		
Nebraska	4	180.50	184.13	29.73	207.01	146.73	100.08	100.93	11.5	104.26	87.46	1.80	1.87	0.15	1.88	1.57		
New York	1	172.20	172.20		172.20	172.20	100.20	100.20		104.26	100.20	1.72	1.72		1.72	1.72		
North Carolina	6	166.82	165.16	13.96	185.00	152.80	94.58	93.80	6.30	104.26	86.00	1.76	1.74	0.08	1.93	1.70		
Ohio	4	180.03	181.75	25.67	205.40	151.20	102.30	104.85	8.97	104.26	89.50	1.76	1.74	0.14	1.93	1.61		
Texas	3	171.68	167.00	11.74	185.04	163.00	99.30	97.00	9.66	104.26	91.00	1.73	1.72	0.05	1.79	1.68		
Tennessee	1	174.48	174.48		174.48	174.48	98.42	98.42		104.26	98.42	1.77	1.77		1.77	1.77		
Utah	2	151.50	151.50	0.71	152.00	151.00	85.50	85.50	2.12	104.26	84.00	1.77	1.77	0.05	1.81	1.74		
Washington	1	161.63	161.63		161.63	161.63	88.82	88.82		104.26	88.82	1.82	1.82		1.82	1.82		
Yukon	5	153.04	154.98	5.27	159.56	146.48	88.44	88.56	0.80	104.26	87.29	1.73	1.73	0.05	1.81	1.68		
M. americanum	89	176.03	174.59	17.71	213.51	143.20	98.84	99.04	8.01	118.00	82.20	1.76	1.77	0.10	1.95	1.57	2.000	
M. pacificus	43	168.20	167.75	14.66	202.77	142.50	84.99	84.28	5.97	104.26	73.08	1.98	1.95	0.14	2.33	1.69	1.000	
Note:

Aggregate M3 length and width measurements, segregated by state/province. Based on published date from Dooley et al. (2019), with additional specimens from this manuscript. Specimens from the first five states listed are assigned to M. pacificus; all other listed specimens are assigned to M. americanum. Measurements are in mm.

Figure 2 Length/width ratios of Mammut M3s.

Symbols in red are M. pacificus. Pink boxes represent the 1σ range for each taxa. Blue boxes represent the 2σ range and black x’s mark the means. The vertical red bar represents the range of possible values for RAM P97.7. Specimen data is based on Dooley et al. (2019) with additional specimens from this manuscript; specimens used are included in Table S1.

Figure 3 Graph plotting Mammut femur length vs midshaft width.

Specimen data is based on Dooley et al. (2019) with addition of the Tualatin mastodon (F-30282) from this manuscript. Direct measurements are in mm.

Another noteworthy Oregon specimen is USNM 4911, an isolated left M2 described as Mammut oregonense by Hay (1926). Dooley et al. (2019) showed that M2 does not differ in any consistent way between M. pacificus and M. americanum, even in L:W ratio. As M. oregonense is only represented by an M2 and no other specimens have ever been referred to this taxon, we concur with assessment of Dooley et al. (2019) that M. oregonense should be considered a nomen dubium, and its use restricted to the holotype.

Washington

Dooley et al. (2019) assigned three specimens from Washington to M. americanum, two mandibles that included m3s and an isolated M3, all from different localities. The isolated right M3 (UWBM 83312) from Jefferson County is a tetralophodont tooth missing large areas of enamel (Fig. 4F). There is little to no wear on lophs 4 and 5, but damage on the first three lophs make it impossible to assess the wear in these areas. The L:W ratio is 1.82, closer to the average of M. americanum (1.76) that to M. pacificus (1.98), but within the known range of values for both taxa (Fig. 2).

Figure 4 Mammut specimens from Washington, USA.

(A, B) Mammut sp. (Manis mastodon) right m3, occlusal view (A), right dentary, medial view (B). (C–E) Mammut sp. mandible UWBM 88099, dorsal (C), left lateral (D), and right lateral (E). (F) Mammut sp. right M3 UWBM 83312, occlusal view. (G, H) Mammut sp. mandible UBMW 14491, dorsal (G), left lateral (H), and right lateral (I). All scales = 5 cm.

UWBM 88099 is a mandible from Lewis County that includes the left m2 and m3, and the right m3 (Figs. 4C–4E). The anterior tip of the mandible is damaged, as are both ascending mandibular rami, which are missing the condyles. While the anterior tip of the mandible is imperfectly preserved, there is no indication of alveoli for mandibular tusks. The L:W ratio of the left m3 is 1.94. This value is just outside of 2σ for M. pacificus (the lowest known value for M. pacificus is 1.95, average 2.26), and close to the mean value for M. americanum (1.89) (Fig. 4).

UWBM 14491 is a complete mandible from Clallam County that includes both left and right m1, m2, and m3 (Figs. 4G–4I). The m1s show heavy wear, the m2s are in moderate wear, and the m3s are not yet fully erupted and show only slight wear on the first lophids. This is most equivalent to Laws (1966) Group XVII or XVIII, indicating and age of 28–30 ± 2 African elephant equivalent years. There are no alveoli for mandibular tusks. The right m3 has a L:W ratio of 2.11. This is well within the known range for M. pacificus and is greater than all but one M. americanum in our dataset (n = 134), but still with the 2σ range of M. americanum (Fig. 5).

Figure 5 Length/width ratios of Mammut m3s, segregated by state/province.

Symbols in blue are M. americanum; in black, Mammut sp. Symbols in red are M. pacificus. Pink boxes represent the 1σ range for each taxa. Blue boxes represent the 2σ range and black x’s mark the means. Specimen data is based on Dooley et al. (2019) with additional specimens from this manuscript; specimens used are included in Table S2.

The Manis mastodon was not included in Dooley et al. (2019). This specimen was discovered during excavation of a holding pond near Sequim, Clallam County, Washington in 1977, and became well known because of a reported bone projectile point embedded on one of the ribs (Gustafson, Gilbow & Daugherty, 1979; Waters et al., 2011), although debate continues over the interpretation of this specimen (Haynes & Huckell, 2016; Waters et al., 2023). Carbon dates from bone collagen yielded an age of approximately 13,800 ybp (Waters et al., 2011). The Manis mastodon has never been fully described or figured, but Gustafson, Gilbow & Daugherty (1979) mention tusk segments up to 2 m in length, suggesting that the individual may have been a male. Field sketches reproduced in Gustafson, Gilbow & Daugherty (1979) indicate the presence of numerous ribs, and at least portions of a forelimb including the scapula, humerus, and ulna. They also figure a heavily worn m2, and mention numerous skull fragments.

A number of elements from the Manis mastodon are on exhibit at the Sequim Museum and Arts in Sequim, including a complete right dentary with an in situ m3 (Figs. 4A, 4B). The mandibular symphysis does not have alveoli for mandibular tusks. The m3 is pentaloph, with wear on all five lophids and heavy wear on the first two. This level of wear is consistent with Laws Group XXII or XXIII, yielding an age of 39–43 ± 2 AEY. The L:W ratio of this tooth is 2.09, within the normal range for M. pacificus and much narrower than typical M. americanum m3s (only two M. americanum specimens out of 134 in our dataset are narrower), but still within the 2σ range of M. americanum (Fig. 5).

The Washington State specimens present an interesting dilemma for taxon identification. All the known specimens lack mandibular tusks, but this absence of tusks is not diagnostic in its own right. One specimen (UWBM 88099) has an m3 that falls just outside the 2σ value of M. pacificus and close to the mean of M. americanum; however, due to the absence of lower tusks and the geographic location of this specimen, we hesitate to definitively refer it to either species. Other Washington specimens have M3/m3 L/W ratios that fall within the 2σ range of both taxa, although generally closer to the averages for M. pacifcus than M. americanum, but lack any definitive characters. Therefore, we refer all Washington specimens to Mammut sp. until such time as unequivocal diagnostic material is reported.

Hidalgo

Multiple elements of a single mastodon are known from Rancholabrean deposits at Ventoquipa, Hidalgo, Mexico (UAHMP-311; Bravo-Cuevas, Morales-García & Cabral-Perdomo, 2015) (Fig. 6). Dooley et al. (2019) included this specimen as M. americanum in their dataset, even though the L:W ratio of the M3 of UAHMP-311 is 1.93, close to the mean for M. pacificus (1.98) and close to the maximum value known for M. americanum (1.95) (Table 1, Fig. 2). Measurements of the m3 of this specimen are now available; it has a L:W ratio of 2.29. This is higher than any specimen of M. americanum in our dataset (maximum = 2.23, n = 134), and is greater than the mean for M. pacificus (2.25) (Table 2, Fig. 5). As both M3 and m3 of UAHMP-311 fall within the known range of M. pacificus, and m3 falls outside the 2σ range of M. americanum, we refer this specimen to M. pacificus.

Figure 6 Mammut pacificus UAHMP-31.

Right M3 (A) and right m3 (B), occlusal view.

Table 2 Aggregate m3 data.

State/Province/Country	n	Mean maximum length	Median maximum length	SD	Max	Min	Mean maximum width	Median maximum width	SD	Max	Min	Mean L/W	Median L/W	SD	Max	Min		
California	23	185.84	187.00	13.54	208.82	159.74	82.86	82.90	6.28	94.03	68.00	2.25	2.25	0.14	2.44	1.95		
Idaho	3	185.93	192.90	19.70	201.20	163.70	82.70	192.90	6.81	90.10	76.70	2.25	2.23	0.12	2.37	2.13		
Hidalgo	1	180	180.00		180.00	180	78.55	78.55		78.6	78.6	2.29	2.29		2.29	2.29		
Alaska	3	167.56	169.79	20.64	187.00	145.9	90.76	92.07	10.95	101.0	79.2	1.85	1.84	0.01	1.85	1.84		
Arizona	1	171.1	171.10		171.10	171.1	82.8	82.80		82.8	82.8	2.07	2.07		2.07	2.07		
Colorado	9	182.44	174.80	11.39	202.40	171.00	95.97	95.70	6.09	106.80	87.80	1.91	1.93	0.17	2.23	1.64		
Florida	23	181.59	183.00	14.35	216.50	155.00	96.12	95.90	5.45	111.60	89.10	1.89	1.93	0.13	2.04	1.63		
Illinois	10	192.87	187.65	19.54	240.78	167.00	102.52	101.50	9.51	121.91	85.00	1.88	1.85	0.06	1.98	1.82		
Indiana	9	185.02	188.50	14.00	202.30	164.00	100.13	99.00	5.02	108.00	91.70	1.85	1.81	0.12	2.04	1.66		
Kansas	2	195.00	195.00	11.31	203.00	187.00	100.74	100.74	6.41	105.27	96.20	1.94	1.94	0.01	1.94	1.93		
Kentucky	9	185.35	182.50	14.85	202.70	165.00	98.62	97.10	7.72	116.50	90.74	1.88	1.85	0.10	2.01	1.73		
Louisiana-Mississippi	14	189.12	188.30	25.72	226.50	113.07	103.58	102.46	7.86	119.05	93.00	1.83	1.88	0.22	2.06	1.17		
Missouri	24	189.65	188.65	14.36	213.00	162.00	98.46	98.85	5.93	109.80	86.00	1.93	1.91	0.06	2.07	1.82		
Nebraska	4	181.18	181.05	3.63	184.70	177.90	100.10	100.20	1.27	101.50	98.50	1.81	1.81	0.05	1.87	1.75		
New Mexico	3	166.33	168.00	12.58	178.00	153.00	89.00	93.00	8.26	94.50	79.50	1.87	1.91	0.08	1.92	1.78		
New York	1	196.70	196.70		196.70	196.70	97.60	97.60		97.60	97.60	2.02	2.02		2.02	2.02		
North Carolina	4	180.45	188.90	18.10	190.60	153.40	91.63	92.15	3.00	94.40	87.80	1.97	2.01	0.15	2.10	1.75		
Ohio	4	191.30	191.20	25.25	222.20	160.60	99.40	101.85	8.72	106.90	87.00	1.92	1.89	0.12	2.08	1.82		
Quebec	1	136.00	136.00		136.00	136.00	79.00	79.00		79.00	79.00	1.72	1.72		1.72	1.72		
Tennessee	1	160.90	160.90		160.90	160.90	90.60	90.60		90.60	90.60	1.78	1.78		1.78	1.78		
Texas	5	188.80	195.00	13.37	200.00	168.00	99.40	100.00	5.08	106.00	93.00	1.90	1.91	0.06	1.95	1.81		
Utah	2	169.50	169.50	0.71	170.00	169.00	82.50	82.50	2.12	84.00	81.00	2.06	2.06	0.04	2.09	2.02		
Virginia	1	165.60	165.60		165.60	165.60	89.50	89.50		89.50	89.50	1.85	1.85		1.85	1.85		
West Virginia	1	177.00	177.00		177.00	177.00	97.00	97.00		97.00	97.00	1.82	1.82		1.82	1.82		
Yukon	2	160.40	160.40	3.75	163.05	157.75	81.88	81.88	0.66	82.34	81.41	1.96	1.96	0.03	1.98	1.94		
M. americanum	133	183.62	184.15	15.70	226.50	136.00	96.83	96.95	7.20	121.91	79.00	1.89	1.91	0.12	2.23	1.17	2.000	
M. pacificus	27	185.63	187.00	13.65	208.82	159.74	82.68	82.41	6.14	94.03	68.00	2.25	2.25	0.13	2.44	1.95	1.000	
Note:

Aggregate m3 length and width measurements, segregated by state/province. Based on published date from Dooley et al. (2019), with additional specimens from this manuscript. Specimens from the first three states listed are assigned to M. pacificus; all other listed specimens are assigned to M. americanum. Measurements are in mm.

Jalisco

A left M3 (LACM 1854) (Fig. 7) was recovered in 1955 from Lago de Chapala, near San Luis Soyatlan, Jalisco, Mexico. A rich fauna from this site includes remains from both Mammuthus and Cuvieronius (Lucas, 2008), but LACM 1854 is the only mammutid element identified thus far. The age of the Lago de Chapala specimens has been problematic, potentially ranging from Rancholabrean to Blancan, but the material from San Luis Soyatlan appears to be Rancholabrean (Lucas, 2008; Rufolo, 1998).

Figure 7 Mammut pacificus LACM 1854 left M3.

Occlusal (A), labial (B) and lingual (C) views.

LACM 1854 is a tetralophodont left M3. Large portions of the tooth, including the entire root area, are encrusted with what appears to be a carbonate or other evaporitic mineral. While there is some damage to the pretrite side of the first loph, the other lophs are undamaged. The lophs are simple, lacking the additional conelets that commonly fill the troughs between lophs in gomphotheriids. There is light to moderate wear on the pretrite side of the first three lophs, with the fourth loph showing only very slight wear. There is a distinct cingulum on the anterior margin, but this does not appear to extend to other portions of the tooth.

The length:width ratio of this tooth is 2.07, well within the known range of M. pacificus (1.69–2.33; mean = 1.98) (Table 1, Fig. 2). No M. americanum M3 in our dataset has such a high L:W value (maximum = 1.95; mean = 1.77), and this value falls outside the 2σ range of M. americanum, justifying the referral of LACM 1854 to M. pacificus.

Alberta

A limited number of mammutid remains are known from Alberta, and were reviewed by Jass & Barron-Ortiz (2017). Nearly all of those records represent isolated specimens recovered from gravel pits in the Edmonton area. The most complete specimen is a partial mandible, RAM P94.16.1 (Fig. 8) from the Apex Galloway Pit, located near Edmonton. The m3 in this specimen has a L:W ratio of 1.90, well outside the 2σ range of M. pacificus and within the known range of M. americanum. Moreover, RAM P94.16.1 has well-developed alveoli for lower tusks, which are unknown in M. pacificus but occur in about 30% of M. americanum mandibles (Green, 2006), regardless of age or sex.

Figure 8 Mammut specimens from Alberta.

(A–C) Mammut americanum mandible RAM P94.16.1 in dorsal (A), anterior (B), and right lateral (C) views. Note the large chin tusk alveolar in (B). (D, E), Mammut pacificus left M3 RAM P.97.7.1 M3 in labial (D) and occlusal (E) views, with a (F) reconstruction of min/max loph widths (blue = min, red = max).

A second specimen, RAM P97.7.1 (Fig. 8), also comes from an Edmonton-area gravel pit (Pit 46). Although damaged, this specimen is identified as a partial left M3 based on the right angle formed by the loph axis and the long axis of the tooth (note: this specimen was reported as an m3 in Jass & Barron-Ortiz (2017)). The tooth includes five lophs, but the first two lophs are damaged, making direct measurement of the maximum tooth width impossible, as in M3 the widest part of the tooth is always at either the first or second loph.

In order to estimate the likely maximum width of RAM P97.7.1, we calculated individual loph widths as a percentage of maximum loph width for 34 Mammut M3s, including 17 specimens each of M. pacificus and M. americanum (Table 3). This enabled us to calculate a range of likely values of the maximum width of a tooth for a given width of the third loph.

Table 3 Mammut M3 loph width.

Specimen	Taxon	County	State/Province	1st loph width	2nd loph width	3rd loph width	4th loph width	5th loph width		1st loph width/widest loph width	2nd loph width/widest loph width	3rd loph width/widest loph width	4th loph width/widest loph width	5th loph width/widest loph width	
Perris mastodon	M. pacificus	Riverside	CA	87	82.3	82.4	60.8	44		1.000	0.946	0.947	0.699	0.506	
SBMNH specimen B	M. pacificus	Santa Barbara	CA	83.9	85.97	81.72	71.18	49.9		0.976	1.000	0.951	0.828	0.580	
SBMNH specimen A	M. pacificus	Santa Barbara	CA	94.88	104.26	102.39	87.02			0.910	1.000	0.982	0.835		
SDSNH 116399	M. pacificus	San Diego	CA	84.46	80.79	77.31	65.35			1.000	0.957	0.915	0.774		
UCMP 1060	M. pacificus	Tuolumne	CA	78.13	75.79	72.87	55.12			1.000	0.970	0.933	0.705		
LACM-HC 87076	M. pacificus	Los Angeles	CA	73.08	72.54	67.97	53.3			1.000	0.993	0.930	0.729		
UCMP 1567	M. pacificus	Tuolumne	CA	78.54	80.04	74.95	57.64			0.981	1.000	0.936	0.720		
UCMP 212936	M. pacificus	Alameda	CA	94.64	95.5	91.05	81.69	60.93		0.991	1.000	0.953	0.855	0.638	
UCMP 36684	M. pacificus	Alameda	CA	77.91	76.18	73.47	66.12	47.31		1.000	0.978	0.943	0.849	0.607	
UCMP 41642	M. pacificus	Sonoma	CA	90	89.36	87.64	71.06			1.000	0.993	0.974	0.790		
UCMP 45265	M. pacificus	Contra Costa	CA	86.33	89.27	87.81	74.53	49.74		0.967	1.000	0.984	0.835	0.557	
UCMP 70139	M. pacificus	Sonoma	CA	86.14	84.35	79.22	66.35			1.000	0.979	0.920	0.770		
WSC 10829	M. pacificus	Riverside	CA	85.2	81.8	80.3	65.9			1.000	0.960	0.942	0.773		
WSC 19730	M. pacificus	Riverside	CA	89.5	89.3	84.2	60.5			1.000	0.998	0.941	0.676		
WSC 22587.1	M. pacificus	Riverside	CA	86.8	84.4	80.9	72.4			1.000	0.972	0.932	0.834		
WSC 9964.7	M. pacificus	Riverside	CA	75.4	74	65.1	46.8			1.000	0.981	0.863	0.621		
WSC 18743	M. pacificus	Riverside	CA	79.97	84.1	73.31	55.69			0.951	1.000	0.872	0.662		
NMC 8060	M. americanum		AK	93.86	94.15	90.39	59.34			0.997	1.000	0.960	0.630		
DMNH 60675	M. americanum	Pitkin	CO	98.3	96.1	87.8	58.4			1.000	0.978	0.893	0.594		
DMNH 69327	M. americanum	Pitkin	CO	99.4	100.2	95.1	75.6			0.992	1.000	0.949	0.754		
DMNH 69331	M. americanum	Pitkin	CO	96.3	98.2	90.4	65.7			0.981	1.000	0.921	0.669		
DMNH 69943	M. americanum	Pitkin	CO	101.2	97.9	95.5	77.3			1.000	0.967	0.944	0.764		
LACM 130386	M. americanum	Bureau	IL	108.07	111.37	102.91	93.1			0.970	1.000	0.924	0.836		
LACM 154685	M. americanum	Allen	IN	83.35	87.85	87.78	62.81			0.949	1.000	0.999	0.715		
ANSP 13309	M. americanum	Boone	KY	96.95	92.68	90.2	68.37			1.000	0.956	0.930	0.705		
ANSP 13310	M. americanum	Boone	KY	86.83	83.1	82.32	65.99			1.000	0.957	0.948	0.760		
LSUMG V-17071	M. americanum	West Feliciana	LA	118	117.7	115	94.8			1.000	0.997	0.975	0.803		
USNM 437571	M. americanum	Dare	NC	96	93	89	78	56		1.000	0.969	0.927	0.813	0.583	
UNSM1642	M. americanum	Dodge	NE	100.9	108.58	102.1	95.45	44.7		0.929	1.000	0.940	0.879		
UNSM2042-69	M. americanum	Nuckolls	NE	93.28	87.2	82.42	57.72			1.000	0.935	0.884	0.619		
UNSM1491	M. americanum	Cass	NE	109.24	110.98	107.5	95.02	56.41		0.984	1.000	0.969	0.856		
UNSM1369	M. americanum	Thurston	NE	86.15	87.46	81.88	70.56	36.15		0.985	1.000	0.936	0.807		
25BJS76	M. americanum	Hickory	MO	107.01	105.25	103.6	81.36			1.000	0.984	0.968	0.760		
NMC 8707	M. americanum		Yukon	86.91	87.29	83.05	56.73			0.996	1.000	0.951	0.650		
									Maximum	1	1	0.999	0.879	0.638	
									Minimum	0.910	0.935	0.863	0.594	0.506	
									Average	0.987	0.984	0.939	0.752	0.579	
Note:

Mammut M3 loph widths. The last five columns describe the width of the given loph divided by the width of the widest loph in each specimen. Bolded fields indicate the widest loph on each tooth. Direct measurements are in mm.

RAM P97.7.1 is 183.69 mm long, while the third loph has a width of 81.27 mm. Using the values of Mammut specimens from Table 3 as a guide yields a range of likely widths for the widest loph, from 82.6–94.1 mm. These values result in a LW ratio between 1.95–2.26 (with an average of 2.12). Reconstructions of RAM P97.7.1 using the estimated maximum and minimum loph widths are shown in Fig. 8. As almost this entire range of values falls outside the 2σ values of M. americanum (1.56–1.96), we refer this specimen to M. pacificus.

Discussion

The presence of M. pacificus in Oregon is consistent with earlier reports of this taxon from northern California, Idaho, and Montana (Dooley et al., 2019; McDonald et al., 2020) (Fig. 9). These records highlight a broad distribution of M. pacificus across the Pacific Northwest and northern Rocky Mountain region of the western United States. At least some of those records (e.g., Montana) pre-date the late Pleistocene and may provide an opportunity to explore further paleobiological questions (e.g., do early records in Montana reflect greater capacity to occupy an array of environmental niches or are they a reflection of earlier Pleistocene environmental perturbations?). The specimens examined in this study do not definitively demonstrate the presence of M. pacificus in Washington even though its presence might be predicted based on biogeographic patterns.

Figure 9 Late Pleistocene distribution map of Mammut pacificus and Mammut americanum.

Based on specimens examined in this article, Karpinski et al. (2020), McDonald et al. (2020), and Dooley et al. (2019). Note that these distributions are approximate and most likely fluctuated with time.

The presence of M. pacificus in Jalisco and Hidalgo is a significant and surprising range extension for this taxon. The Mexican record represents the southernmost occurrences of Rancholabrean M. pacificus, inhabiting areas that now are part of west-central and central Mexico. Given that Texas and New Mexico specimens are assignable with some confidence to M. americanum (Dooley et al., 2019), it seems that the range boundary near the southern margins of the distribution for these two species lay somewhere in northern Mexico during the Late Pleistocene.

Karpinski et al. (2023) reported the presence of mitochondrial genome material consistent with M. americanum from American Falls, Idaho, a site that has produced specimens referred to M. pacificus based on morphology (Dooley et al., 2019). Here we add Alberta to Idaho as states/provinces that have produced specimens of both M. americanum and M. pacificus, although it is unclear if these taxa were present contemporaneously in each location. Nearly all mastodon specimens from Alberta, and much of the record of megafauna of Alberta, were recovered as part of industrial gravel extraction (Jass & Barron-Ortiz, 2017). Precise contextual data are not available for most specimens, inhibiting our ability to temporally relate individual specimens from the region that lack C-14 data or exceed the capabilities of radiocarbon dating. Direct dates on the Alberta specimens discussed here are either infinite (P97.7.1; >41,100 14C yr BP; Metcalfe et al., 2016) or close to infinite and in need of re-evaluation (P94.16.1; 40,700 ± 3,000 14C yr BP; Jass & Barron-Ortiz, 2017). Although our ability to relate the specimens in time is somewhat challenged, that does not diminish the significance of the observation of both taxa in the same geographic region.

The eastern Montana M. pacificus specimen reported by McDonald et al. (2020) lies far to the east of the Alberta occurrence of M. americanum, suggesting that the ranges of these taxa may have overlapped significantly in the northern Great Plains or that the range boundaries may have fluctuated over time. Although limited temporal control leaves that question presently unresolved, we note that the record of both M. americanum and M. pacificus in Alberta points to further complexity in movement of taxa through the interior of northern North America during the Pleistocene, as noted by Karpinski et al. (2020). South-to-north dispersals through Alberta may have been influenced by population sources from both sides of the Rocky Mountains.

These data are of particular interest when considered in the context of mitochondrial genome data for Mammut described by Karpinski et al. (2020). They found a high level of endemism in Mammut populations in all regions they studied except in Alberta, where there were specimens with phylogenetic affinities to Missouri, Alaska, and Mexico. The single specimen with genetic affinities to Mexican specimens was RAM P.97.7.1, and is the same tooth that we have morphologically identified as M. pacificus. This suggests that the “Clade M” of Karpinski et al. (2020), which included RAM P97.7.1 and the Mexican specimens, may represent M. pacificus, while their clades Y, G, L, N, and A, taken together, represent M. americanum. According to Karpinski et al. (2020), Clade M diverged from the other clades at 3.03 Ma, indicating that M. pacificus and M. americanum likely diverged from each other sometime in the Pliocene. Examination of Early Pleistocene and Pliocene mammutids along with better age constraints on known specimens should help illuminate the nature of the divergence of these taxa as well as the biogeographic changes that have taken place in North America during the Neogene.

Supplemental Information

Supplemental Information 1 Raw data.

Supplemental Information 2 Upper M3 specimen data.

Supplemental Information 3 Lower m3 data.

The authors would like to thank the Tualatin Pubic Library, the Sequim Museum and Arts, the Natural History Museum of Los Angeles County, Royal Alberta Museum, Museo de Paleontología, Universidad Autónoma del Estado de Hidalgo, and the Burke Museum of Natural History and Culture for access to specimens in their care. We would like to thank Greg McDonald for his useful comments, and Melissa Pardi, Michael Cherney, an anonymous reviewer, and editor Steven Heritage for their helpful feedback, which greatly improved this manuscript. We would also like to thank Lauren Loustaunau and Erik Ozolins for additional copyediting assistance.

Institutional Abbreviations

F- Tualatin Public Library, Tualatin, Oregon, USA

DMNH Denver Museum of Natural History, Denver, Colorado, USA

LACM Natural History Museum of Los Angeles County, Los Angeles, California, USA

RAM Royal Alberta Museum, Edmonton, Alberta, Canada

LSUMG Louisiana State University Museum of Natural Science, Baton Rouge, Louisiana, USA

NMC Canadian Museum of Nature, Ottawa, Ontario, Canada

SBMNH Santa Barbara Museum of Natural History, Santa Barbara, California, USA

SDSNH San Diego Natural History Museum, San Diego, California, USA

UAHMP Museo de Paleontología, Universidad Autónoma del Estado de Hidalgo, México

UCMP University of California, Berkeley Museum of Paleontology, Berkeley, California, USA

USNM United States National Museum of Natural History, Washington DC, USA

UWBM University of Washington Burke Museum Seattle, Washington, USA

WSC Western Science Center, Hemet, California, USA

Additional Information and Declarations

Competing Interests

Author Contributions

Data Availability

Eric Scott is employed by Cogstone Resource Management. All other authors declare that they have no competing interests.

Alton C. Dooley Jr conceived and designed the experiments, performed the experiments, analyzed the data, prepared figures and/or tables, authored or reviewed drafts of the article, and approved the final draft.

Chris Widga conceived and designed the experiments, performed the experiments, analyzed the data, authored or reviewed drafts of the article, and approved the final draft.

Brittney E. Stoneburg analyzed the data, prepared figures and/or tables, authored or reviewed drafts of the article, and approved the final draft.

Christopher Jass analyzed the data, prepared figures and/or tables, authored or reviewed drafts of the article, and approved the final draft.

Victor M. Bravo-Cuevas analyzed the data, prepared figures and/or tables, authored or reviewed drafts of the article, and approved the final draft.

Andrew Boehm analyzed the data, prepared figures and/or tables, authored or reviewed drafts of the article, and approved the final draft.

Eric Scott conceived and designed the experiments, performed the experiments, analyzed the data, authored or reviewed drafts of the article, and approved the final draft.

Andrew T. McDonald conceived and designed the experiments, analyzed the data, authored or reviewed drafts of the article, and approved the final draft.

Mark Volmut analyzed the data, prepared figures and/or tables, authored or reviewed drafts of the article, and approved the final draft.

The following information was supplied regarding data availability:

The raw measurements of associated molars are available in the Supplemental Files.

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
