# Peer review of "Re-evaluation of mastodon material from Oregon and Washington, USA, Alberta, Canada, and Hidalgo and Jalisco, Mexico"

_PeerJ, doi:10.7717/peerj.18848_

## Round 0.1 · original submission · Major Revisions

All three of the reviewers recommend Major Revisions with which I agree. To move forward, please address all comments from each of them. In the Materials & Methods section, please explain your data collection methods in addition to citing the two previous manuscripts. In your revision, it will be important to include appropriate statistical tests given the range overlaps in L:W ratios. In Figures 2 & 4, please indicate that the red and blue shaded regions are the one standard deviation ranges.

·

Basic reporting

The article is written in clear English throughout. There are a small handful of a typos or sections needing additional clarification which I note line-by-line. Sufficient background information and context are provided. A strength of this study is how it relates morphology to previous findings that used ancient DNA to identify distinctive lineages. There are some minor revisions that are needed for the table and figure captions. Most of the raw data are shared; however, the paper makes reference to data not presented as part of an analysis, I detail this below. The paper is self-contained with relevant results to stated hypotheses.

Line 37: "is" should be "it"

Line 87: I'm not sure what is meant by "a nearly ubiquitous part of the Late Pleistocene fauna" in this opening statement.

Line 142: The values for P3s between the two species are not given in this manuscript. If they are indicating they are more similar to M. pacificus, this should be shown/provided.

Line 175: "that" should be "than"

Line 204-205: It's not clear why Table 1 contains radiocarbon data from Gilmour et al. but Waters et al. is not included in any more detail.

Line 327: Clarity would be improved if the opening sentence of this paragraph was less ambiguous on what "these data" refers to. It's not clear if it is referring to the previous paragraph or the manuscript overall.

Figures 2 and 4: please indicate in the figure caption what the shaded areas signify.

Figure 8: to facilitate the inclusion of the data in this paper in online repositories, it would be helpful if the authors included a supplement that provided the site names, locations, and ages (if known).

Table 1: the caption could be improved by explicitly including the name of the locality these dates refer to, which would enable the reader to fully interpret the purpose of the table on its own.
In general, the table captions, as written, do not contain enough detail to permit stand-alone understanding of what is being shown.

Tables 2 and 3: there are many entries where the value is showing as ####. The supplemental file provided also has this issue. What are the values here?

Experimental design

This study represents original primary research that fits the scope of the journal. It is clear how this work fills in previous gaps of knowledge of mastodon taxonomy and phylogeography. The rigor of the statistical methods could be strengthened; however, I do not see this as a critical flaw of the study. I elaborate on this below. Improving the rigor of the analyses will strengthen the reproducibility of this work.

Materials and Methods: the methods could be more robust if it was more explicitly stated, in a quantitative way, how L:W ratios were judged. It's clear that when values are outside of the overall range for M. americanum, then the individual in question is M. pacificus. However, when an individual specimen has a ratio that falls where there is overlap between the two species, what is the criteria? If the authors are using confidence intervals in their assessment, and I recommend they should, this could be explicitly stated in the methods.

Line 142: This is an example of what I was referring to in my comment on the methods. How was it determined that these were more similar? Does it fall outside of the range of M. americanum? Is it within the 95% CI for M. pacificus but not M. americanum? A quantitative argument would be stronger.

Lines 174-176: this is an example where using confidence intervals would be useful

Lines 190-192: given there is overlap, albeit small, a confidence interval would be more appropriate.

Lines 216-218: given there is overlap, use of CIs would make a more robust argument.

Validity of the findings

Most of the data that the findings are based on are reported, save for the comment I made in reference to Line 142. The conclusions are appropriately stated and connected to the original questions being investigated.

Line 157: How does it compare to M. americanum values?

·

Basic reporting

In the introduction, the author’s do a good job of highlighting the potential contributions their findings could make to the broader understanding of mastodon paleobiology. In two instances, I recommend more recent references to replace/augment the references included:

1) In lines 199-204, the authors introduce the Manis mastodon. Although not pertinent to the conclusions of this manuscript, the authors cite as background information for the Manis mastodon original work by Gustafson (1979), subsequent work by Waters et al. (2011), and a refutation of the Waters et al. conclusion contributed by Haynes and Huckell (2016). In order to provide up-to-date treatment of the Manis mastodon scholarship, the authors should include the following recent publication:
Waters, M. R., Newell, Z. A., Fisher, D. C., McDonald, H. G., Han, J., Moreno, M., & Robbins, A. (2023). Late Pleistocene osseous projectile point from the Manis site, Washington—Mastodon hunting in the Pacific Northwest 13,900 years ago. Science Advances, 9(5), eade9068.
2) Peecook et al. (2022) is a conference abstract. The study has since been published in Quaternary International. If the authors keep the reference to the abstract, the spelling of the lead author’s name (Peecook rather than Peecock) should be corrected. The published research paper that reports the relevant findings of Peecook et al. (2022) is:
Karpinski, E., Widga, C., Boehm, A. R., Peecook, B. R., Kuch, M., Murchie, T. J., & Poinar, H. N. (2023). Mastodon mitochondrial genomes from American falls, Idaho. Quaternary International, 668, 1-6.
The organization of the article is easy to follow and conforms to convention. However, the figures and tables would be improved by some additional efforts to improve clarity.

The specimen photos show the relevant morphology of dental features discussed in the manuscript. However, it would be useful to have the left femur of the Tualatin mastodon figured as it is described and relevant to the identification of Tualatin as M. pacificus. On the other hand, some figure elements including lateral views of mandibles and molars in 3B (Manis mastodon), 3D,E (UWBM 88099), 3H,I (UBMW 14491), 6B,C (LACM 1854), 7C (RAM P94.16.1), and 7E (RAM P.97.7.1) appear to be superfluous for the purposes of this study and their omission would make room for larger images of relevant views (current 3A,C,F,G, 6A, 7A,B,D). Because only length and width measurements of molars and absence/presence of mandibular tusks (and length/width of the femur for Tualatin), only images showing those features are necessary.
The plots of L:W ratios for specimens according to state/province (Figs. 2 and 4) are useful in that they can easily be compared to charts previously published based on this data. The added overlain boxes are helpful to visualize the normal range for each recognized taxon, but the caption needs to clarify what the boxes represent (mean +/- 1 SD?). For clarity, please add explanation of how states/provinces were ordered along the x-axis. Also, for ease of reading, I recommend labeling the x-axis. Other minor issues are noted in the annotated pdf.
Table 1 is not clearly necessary. If included here (rather than only referencing the original Gilmour et al. publication), then inclusion should be specifically justified.
Tables 2 and 3 appear to be slight modifications of tables 4 and 5 from Dooley et al. (2019). It is unclear if they are intended here as in-text tables here or as supplementary data. Although I think it is important to provide such measurement data, the organization of these tables is not explicitly relevant to this particular study. A smaller table with only data relevant to the taxonomic designations made in this manuscript (L:W means, maxs, mins, standard deviations, and region information for M3s and m3s of M. americanum and M. pacificus) would be very helpful. I have made a number of suggestions for clarity in the annotated pdf.
Table 4 is important data to include, but to make the width estimate more reproducible, the specific approach to calculating estimated maximum widths should be spelled out and/or specific estimates based on each specimen’s data should be included as a column. Specific suggestions to make the table more clear and readable are included in the annotated pdf.
Overall, the manuscript is written clearly. In the annotated pdf, I have noted a few instances where I think the authors should reconsider wording for clarity.

Experimental design

This study by Dooley Jr. et al. follows a 2019 publication by Dooley Jr. et al. that erected a new taxon, Mammut pacificus, for late Pleistocene mastodons in California as well as specimens in Idaho, as a sister group to the long recognized North American taxon, M. americanum. A subsequent contribution from MacDonald et al. (2020) proposed expanding the range of M. pacificus to include a single specimen from Montana. The current manuscript proposes additional range extensions for M. pacificus based on evaluations of specimens from Washington, Oregon, Mexico, and Alberta and in some regions proposes range overlap between M. pacificus and the more widespread M. Americanum (though the authors defer interpretations regarding potential niche partitioning until the accumulation of more well-dated specimens makes it possible to conclude that the two taxa occupied the same spaces simultaneously). The focus of this work (mastodon taxonomy and paleobiogeography) has important relevance for the paleobiological understanding of North American mastodons and is of particular interest considering regional variations in mastodon morphology and genetic evidence for regional endemism (matrilineal, at least).
Although the conclusions drawn from the data presented here are not necessarily in error, here I identify what I think are flaws in the experimental design that should be addressed prior to publication.
The strategy for taxonomic resolution performed by Dooley Jr. et al. (2019) implicitly assumed geographically distinct populations--Not unlike recognition of island populations as distinct taxa, “Pacific” populations from California were recognized as being generally smaller with narrower teeth and identified as a distinct evolutionary lineage. So long as the taxa are recognized as having distinct geographic ranges, morphological overlap is no complication. In that case, if from a specimen is from California, it is likely M. pacificus even if its morphology overlaps that of the eastern taxa, M. americanum. Meanwhile, if not from California, a specimen is treated as likely M. americanum and will only be considered M. pacificus if consistent with “reasonable” range variations for a California-centric population and has morphology clearly outside of the norm for M. americanum. However, when taxa overlap both morphologically and geographically, taxonomic identification of individual specimens becomes complicated. In that case, specimens with characters falling within the range of morphological overlap can be either taxon no matter where they were found and specimens near the margins of the overlapping morphological and geographic ranges can be seen as either extending the morphological disparity of the one taxon or extending the geographic disparity of the other. Which character takes precedence could be informed by independent knowledge of geographic flexibility and morphological plasticity, but without such knowledge, either interpretation is equally valid. Furthermore, acknowledging geographic overlap calls into question the basis for some of the taxonomic decisions that went into constructing the original dataset. If the ranges are not distinct, might specimens from California with characters within or near the documented overlap actually be M. americanum and might complementary specimens from outside of California actually be M. pacificus? If we consider specimens outside of California as potentially M. pacificus, what criteria prohibit interpreting the narrowest-molar specimens from CO, MO, IN, and OH (which all have L:W ratios closer to the average of M. pacificus than to the average of M. americanum) as M. pacificus, thus extending the “known” range of M. pacificus to include the Great Lakes region?
The taxonomic interpretations in this study appear to be based on the dataset of measurements provided in Dooley et al. (2019) (this is not clarified, and one of my comments in the annotated pdf requests that clarification be added). As noted by the Dooley et al. (2019), all of the diagnostic features for M. pacificus (narrower third molars, absent mandibular tusks, 6 sacral vertebrae, robust femora, smaller tusks) also occur in M. americanum. Thus, discriminating between these taxa either requires the presence of features that are convincingly outside of the overlapping morphospace or requires a combination of features that together provide convincing evidence as to its identification. In order to justify a geographic range extension for M. pacificus while acknowledging range overlap with M. americium, a tooth-only specimen needs “unambiguous” molar L:W (a value that would be identified as M. pacificus regardless of whether it came from inside or outside the known geographic range). In that case, as long as the metric is accepted as independently diagnostic for the M. pacificus, it could be used to extend the known geographic range of the taxon. However, extending the geographic range of the “geographically-defined” taxon M. pacificus requires conclusive morphological evidence.
If my assessment of flaws in this manuscript is justified, the authors should reconsider proposing M. pacificus range extensions to Washington and Alberta, considering that those specimens identified as M. pacificus could justifiably be attributed to M. americanum on the basis of their morphologies being within or near the known variability for that taxon and that these regions are part of the already accepted range for that taxon (in fact, this study supports the presence of M. americanum in these regions). In addition, subsequent nuanced taxonomic attributions that recognize the complexity of shifting and potentially overlapping ranges would benefit from revising the dataset to function less as observations of variation in two populations that are assumed to be isolated from each other (as was implied by the Dooley et al. treatment in 2019) and more as a tool for diagnosing specimens based on morphology independent of geographic origin (as is implied by the approach of this manuscript). In that case, geographic range would become apparent based on results of morphological analyses (as is being proposed in this manuscript) rather than be a starting assumption (as was implied in Dooley Jr. et al (2019)). To revise the dataset accordingly would require purging any specimens that could not be conclusively identified independent of geography (e.g., tooth-only specimens with dimensions that fall within or near the morphological overlap).
Perhaps out of necessity, or perhaps by design, the authors do not address “intermediate” dental morphologies between the eastern and western Mammut populations (apparent in Figs. 2 and 4). This may be considered a minor issue, but it seems worth consideration by the authors and elaboration regarding what they think is going on. In lines 216-218, they state that the Manis mastodon molar L:W (2.09) is “much narrower than typical M. americanum m3s”, but this disregards regional trends within the dataset. L:W of 2.09 is within 1 SD (0.07) of the average (2.04) for specimens west of the Rockies in those designated in Dooley et al. (2019) as M. americanum. If any of the other western specimens in the M. pacificus dataset (for example, specimens from California) are in fact M. americanum then the “West of the Rockies” average could go up.

Validity of the findings

Considering potential problems I flag in the experimental design--in the context of this study, the identifications of M. pacificus in Oregon, Hidalgo, and Jalisco are consistent with treating only “unambiguous” identifications as rationale for extending the known range for the taxon (this still assumes that “M. pacificus” specimens in reference dataset from California do not include narrow-toothed M. americanum misidentified due solely to their presence in California). However, to avoid using questionably attributed specimens to inform the “known” range of the M. pacificus, the two specimens proposed as extending the range of M. pacificus into Washington (their dimensions are within recognized range of M. americanum) and the one specimen proposed as extending the range to Alberta (the “reconstructed” measurement falls just outside of the overlapping morphospace) should not be the basis for amending the known geographic range and their morphologies should not be added to the reference dataset for M. pacificus. These three specimens can be considered candidates for M. pacificus in Washington and Alberta, but are also consistent with demonstrating some of the morphological disparity within M. americanum (after all, this paper asserts the presence of M. americanum in both Washington and Alberta).
If the dataset used for taxonomic identification of specimens is the same as that published with Dooley Jr. et al. (2019), then that needs to be explicitly stated. If, however, the dataset has been updated since that original publication, then the changes to the dataset should be noted or the new dataset should be included in supplementary data.

Additional comments

Starting with their 2019 contribution, Dooley Jr. and colleagues have been making contributions our understanding of North American mastodon taxonomy. As demonstrated by the morphological evidence compiled by Dooley Jr. et al. (2019) and the genetic evidence presented by Karpinski et al. (2020), the phylogeny of North American mastodons is almost certainly better accommodated by multiple taxa rather than a single continent-wide, highly variable Mammut americanum. The morphological characters recognized by Dooley Jr. et al. (2019) as the basis for describing a new taxon are reasonable and there is no expectation that closely related species will not have overlapping morphologies. However, discrimination between them, especially for the purpose of determining geographic range requires convincing diagnostic evidence independent of geography--either the presence of at least one unambiguous character or a suite of characters that together overwhelm any doubt. In many cases, we just can’t know, as Dooley et al. recognize with multiple attributions to Mammut sp.

Reviewer 3 ·

Basic reporting

The English is fine and clear.

Very insufficient background. The reader is thrown in with an assumption that they know exactly what previous authors found. Related to this, the present authors provide no information or explanation of their own chosen methods.

Literature is OK. Figures and Tables OK, all data presented.

There are no clear hypotheses stated, whereas one potential one seems very obvious to me (see section 3 below).

Experimental design

Yes, within aims and scope of PeerJ.

The research question is pretty well defined.

The investigation is insufficiently rigorous. The authors show little appreciation of individual variation, and none of the necessity for statistical tests. They assert that a specimen belongs to a certain species on the basis of marginal metrical differences, with no statistical testing. Many examples given in section 4 below.

The Methods are not described at all.

Validity of the findings

Little on impact or novelty.

The data are all provided and seem OK; the treatment of them is not statistically robust, as indicated above.

Because of the lack of statistical control, the conclusions do not convincingly follow from the data. Moreover, I feel there is a significant missed opportunity in this paper, even with the data they have to hand. The authors seem to take the presence of two distinct mastodon species as holy writ, yet it was only proposed 5 years ago, and the morphological distinctions, as they describe them, seem rather weak. What if it’s just individual or population-level variation within a single species? Contained within this paper are strands of evidence that, taken together, might support the validity of the two species, but the authors do not bring that out because they assume it from the start. This paper would have much more interest and value if these topics were fully discussed. Further details in (4) below.

Additional comments

This is an interesting paper but needs significant improvement in three key areas:

1. More background. The reader is thrown in with an assumption that they know exactly what Dooley et al and Karpinsky et al found. Related to this, the authors provide no information or explanation of their chosen methods.

2. An appreciation of individual variation, and application of statistical tests, before asserting that a specimen differing by a small metrical degree from another must be a different species (see examples below).

3. A deeper Discussion. I feel there is a missed opportunity in this paper to discuss and critically assess the basis for ‘M. pacificus’. The authors seem to take the presence of two distinct mastodon species as holy writ, yet it was only proposed 5 years ago, and the morphological distinctions, as they describe them, seem rather weak. What if it’s just individual or population-level variation within a single species? I’m not saying that is the case but it’s surely worthy of discussion? Contained within this paper are strands of evidence that, taken together, might support the validity of pacificus, but the authors do not bring that out because they assume it from the start. The combination of a number of specimens edging toward ‘pacificus’ morphology, even if individually they lack statistical significance as I point out below, may be meaningful if they have a coherent geographical range. The fact that Mexico and Alberta are in the most deeply divided clade in mtDNA, is consistent with this interpretation as far as it goes (more data needed). I remain puzzled that two species divided 3 million years ago are identical in dentition except for L/W proportion of M3 (that is well-known to vary by at least this amount in other proboscidean species, e.g. Mammuthus trogontherii: Lister 2022 QSR 294, 107693). Hence personally I reserve jusgment about these two ‘species’. But the authors are entitled to support their belief in a pacificus species if they consider all the available data together. This paper would have much more interest and value if these topics were fully discussed.

Detailed comments (not necessarily ‘minor’!)

86 et seq.: the Introduction is too short. It should describe the original basis for the erection of M. pacificus and in particular, its supposed diagnostic features that are key to the current paper. The reader should not be expected to be fully conversant with the Dooley paper.

89: ‘surprising’ twice in one line

109: ‘Methods’ section is inadequate – in fact there are no methods described or explained. All we can deduce is that that length and width of third molars were measured – is that all? And to what purpose?

142: We are immediately diving into ‘L/W ratios’ without explanation – this was (presumably) one of Dooley’s distinguishing features (the only one??) but the authors have not even explained which of the two species has the higher or lower value, what those values are, whether they are fully distinct or overlap, etc.

147: where is Tualatin (apart from being in Oregon)?

155: Stating that the Tualatin value is outside the range of americanum in the authors’ dataset sounds potentially circular – how were the molars assigned to species while the ranges were being established? Again, is this the only feature by which the species can be distinguished – nothing about occlusal morphology for example? This seems especially problematic since the ranges of the two species overlap substantially (1.59-1.95 vs 1.69-2.33). A value of 2.07 seems to me very dubiously assigned to either of these unless those ranges were based on very large sample sizes - it is only 0.12 (6%) above the americanum upper bound. The authors have to conduct a significance test (a one specimen vs a sample t-test would do) – they need to show that vs one of the ranges the specimen is signficantly different and against the other, not.

156-158: The femur is a little more plausible, if size is in fact a distinguishing feature (again, the authors have not mentioned this), though Dooley’s sample is small (6 femora assigned to pacificus), and proboscidean body size is both sexually dimorphic and notoriously flexible. So I would say this is a suggestion rather than an ‘indication’. Also, I assume the femur is a fully fused adult (this needs to be stated).

If small size and a different L/W proportion of molars are the only two features, have either the authors or Dooley constructed an allometric analysis to check that the proportion difference is not simply an allometric effect of small sized individuals (or local populations) within M. americanum?

176: So, the conclusion is…. ? Presumably that the speicmen cannot be allocated to ‘species’. This needs to be stated (same for each of these specimens).

181: are mandibular tusks mentioned just for general interest, or is this a putative distinguishing feature between the species?

182-3: It’s very unlikely (with reference to comments above about statistics) that 1.94 is significantly different from a range starting at 1.95 with n=29. Again, it needs to be tested, and some conclusion on the status of the tooth given, rather than leaving it with a vague implication that it’s most likely to be pacificus.

191-2: Same again, a vague implication that this is pacificus, but in fact it falls within the ranegs of both species.

194-7: Given all the above, I’m afraid I find these conclusions unconvincing.

215-6: Can Laws’ age categories for African elephants be extended to mastodons with a significantly different pattern of dental replacement (more primitive – frequently 3 molars in wear at a time, that is never seen in elephants)? It also assumes similar longevity – is this known? It’s fair enough to conclude it’s a middle-aged adult, but I wouldn’t put a number on it.

216-9: again, statistical testing needed.

219: So finally we learn that absence of mandibular tusks is thought to be a distinguishing feature of M. pacificum. This needs to be stated and discussed earlier in the ms. How consistent is this in the two species? How many mandibles of each species have both m3 and the mandibular symphysis to demonstrate the correlation? Maybe this information is in Dooley’s paper but readers of the current ms shouldn’t be expected to be fully conversant with that, and the authors should give their own opinion on how reliable these features are. I have seen mandibles of M. americanum lacking mandibular tusks.

227-228: This seems more reasonable (that 2.29 is statistically above the known range); even so, statistical demonstration is advisable. Good to see n mentioned; it would be useful to do this throughout, when citing ranges or means.

249-52 – again, this depends on n – it may well be good with a sample size of 89, but this needs to be stated or preferably demonstrated.

260: Once again on statistics, what’s the sample size of pacificus synphyses to demonstrate 0% tusk frequency?

271-4: this is a good trick.

284: see request on Fig. 8. One has to have a map of US states in one’s head to see why this is ‘consistent’.

301-3: The reader cannot appraise this without some summary of what Peecock et al found. Is there a reciprocal monophyly in mtDNA between specimens morphologically referred to each species? What is the explanation for the disjunct in the American Falls example? This seems rather critical. It’s very common for mtDNA and morphology not to match within and between closely related species (because of lineage sorting among other reasons), and the finding of the ‘wrong’ DNA does NOT necessarily mean that both species are present as the authors seem to imply.

305: While I agree with the authors that we can rarely be absolutely certain that two species were present contemporaneously, I would like to see them consider the alternative explanation, that the ‘americanum/pacificus’ divide, based, it seems to me, on rather weak morphological evidence, might be intraspecific variation within one species? In some places one morph might be dominant, in others both might occur (as part of a single interbreeding population).

323: I agree on potential complexity of movement of populations.

327-339: I agree, this is a plausible interpretation and helps shore up the likelihood of these being separate species. But see my introductory recommendations.

Fig. 8: I would suggest labelling relevant states with their two-letter abbreviations, and keying in the caption. Not all readers outside the US will know where Oregon, Montana etc are (sadly). Also, has every dot on the map been verified with molar L/W ratios?

---

## Round 0.2 · Minor Revisions

Hi Brittney,

In its current iteration, it’s my opinion that the PeerJ should accept your manuscript provided that you make several minor revisions. Please do the following:

1. In the co-author list, please correct the superscript for Stoneburg.

2. In the abstract, please remove the comma in this part of the sentence:
"margins of each species, in order to determine range" ( -> "each species in order")

3. In the abstract, please correct the typo:
"present in the same place, is may suggest" ( is -> it )

4. In the introduction, please do not use 'surprising' twice in this sentence:
"it is perhaps surprising that recent research has revealed surprising new"
Perhaps remove the second instance so it reads:
"it is perhaps surprising that recent research has revealed new"

5. In the introduction, please tone down the choice of the word "established", as pointed out by one of your reviewers:
"Dooley et al. (2019) established M. pacificus based primarily on"
Perhaps you could substitute: ( established -> described )

6. In the introduction, please make two changes to this sentence:
"Most of these represent considerable range extensions for M. pacificus, and indicate a more"
( "Most" -> "Some" ) & ( "indicate" -> "may indicate" )

7. In the methods section, please change from plural to singular in this sentence:
"ratios of mastodon third molars follows Dooley" ( follows -> follow )

8. In the methods section, please change this full sentence as suggested by one of your reviewers:
"Methods for assessing the width of incomplete molars is described with the relevant specimen."
->
"Methods for assessing the width of the incomplete molar from Alberta (RAM P97.7.1) are described with that specimen."

9. In the methods section, please remove the comma in this sentence:
"anomalous if assigned to M. americanum, and hypothesized"

10. The word "database" is used 4 times throughout the manuscript (the first use is in the methods section). Please change all of these occurrences to "dataset".

11. In the methods section, the first time that the sigma character is used, please find a way to state that it means 'standard deviation'. This is obvious, but in my opinion, it is necessary to define.
"the M3 or m3 falls outside of 2σ for"
Perhaps: "the M3 or m3 falls outside of 2σ (standard deviations) for"

12. In the Washington subsection (under Results), in the last sentence, please add a comma after ‘therefore’.
"Therefore we refer to the" ( Therefore -> Therefore, )

13. In the acknowledgments section, please thank your reviewers (by name, if available) for their time and efforts -- even if some of them have asked to remain anonymous.

14. In the Methods section, you stated:
"and the specimens shares an additional character associated with M. pacificus as described in the diagnosis presented in Dooley et al. 2019, with no characters unique to M. americanum."

I think you are referring to traits in the Diagnosis subsection from Dooley et al. 2019, which reads:

"Diagnosis
A species of Mammut differing from Mammut americanum in the following characteristics: M3/m3 significantly narrower relative to length; six fused sacral vertebrae in later ontogenetic stages (usually five in M. americanum, with a range of four to six); femur with a greater midshaft diameter relative to length; absence of mandibular tusks and associated alveoli (variably present in M. americanum); smaller basal diameter of tusks in males for a given age."

In my opinion, it’s necessary to add a sentence or two (at the end of the Methods section of the current manuscript) that restate the definitions of the distinguishing characters that you used. You can simply re-word the relevant text from Dooley et al. 2019. But, please be careful not to self-plagiarize (the next round of copy editing will return this manuscript if the wording is too similar to any previous publication).

--

Nice work. Thanks for this contribution and for all of the time and effort that your team put into it.

All my best,
Steven

·

Basic reporting

As stated in my initial review, the article is written in clear English throughout. I found only a small number of minor typos upon my seconds review, and I note these line-by-line. Sufficient background information and context are provided. As noted before, a strength of this study is how it relates morphology to previous findings that used ancient DNA to identify distinctive lineages. The authors have addressed the minor revisions that were needed for the table and figure captions. The authors have included all data necessary to review the study. The paper is self-contained with relevant results to stated hypotheses.

Specific feedback:
Line 4: correct the superscript for Stoneburg
Line 29: remove "in"
Line 243: insert comma after "Therefore"

Experimental design

As stated in my initial review, this study represents original primary research that fits the scope of the journal. It is clear how this work fills in previous gaps of knowledge of mastodon taxonomy and phylogeography. Following revisions, I find that the rigor of the statistical methods have been sufficiently strengthened following the recommendations I gave in my initial review.

Specific feedback
Lines 299-300: it's not obvious what was done with these data to make this estimate. Please provide more explanation. Apologies for not noting this on the initial round of reviews; however, I do not feel that any new analysis is required here, nor do I anticipate it being complicated enough to warrant additional review. I just think more details on how this was done should be included.
Line 303: it is not clear how these values were used as a guide (see previous).

Validity of the findings

I find the findings to be consistent with the data presented in light of the revised methods.

·

Basic reporting

The revised manuscript from Dooley Jr., et al., Re-evaluation of mastodon material from Oregon and Washington, USA, Alberta, Canada, and Hidalgo and Jalisco, Mexico
includes notable additions and revisions that I think are significant improvements on the original submission, and I applaud the authors for their efforts to address the issues flagged in the original manuscript. The changes that stand out to me include the following:
1) Femur of F-30282 added to Figure 1
2) Graph plotting femur length/width that includes F-30282 added (new Fig 3)
3) Table of previously reported radiocarbon data for F-30282 removed
4) Methods section revised to improve clarity:
a. rationale for taxonomic determinations added
b. details added for approach to estimating max. width of X
5) A more conservative approach to the taxonomic interpretations
6) Plots for L:W of m3/M3 (Figs 2 and 5 in revised manuscript) were streamlined for clarity

This version of the manuscript nevertheless still has a few problematic issues that I recommend be addressed before publication. Minor suggestions are included as annotations in the PDF. In addition, I have the following substantial comment:

I do not think the bases for calling the incomplete M3 from Alberta (RAM P97.7.1) M. pacificus and the partial mandible from Washington (UWBM 88099) M. americanum are sufficiently solid to support the claim of overlapping presence of both taxa in each region. Discussing proposed identifications for these with a degree of likelihood is certainly appropriate, but neither represents a particularly strong case. I think that interpretations of this supposed overlap should be either avoided or sufficiently treated as speculation. I develop my critiques further in comments on Experimental Design.

Experimental design

In the methods of this revised manuscript, the authors establish an explicit, more rigorous rationale (quantitative and qualitative) for assigning specimens to M. pacificus. This inspires a bit more confidence in the objectivity and reproducibility of their taxonomic assignments, so I see it as a significant improvement on the first version of the manuscript. Application of these criteria to each specimen they examined appears to have resulted in a few revised interpretations (specimens called M. pacificus in the original submission that are in the revision called M. sp.). I make a few suggestions in the annotated PDF that I think are necessary to carry these changes through the document consistently.

However, a somewhat complex and fundamental issue remains unresolved in this revision. It is still unclear to me how the authors intend to consider geography in their taxonomic interpretations. If, as I understand to be the case, they are attempting to make taxonomic assignments independent of geographic associations, then it is not irrelevant or inconsequential that geographic associations were treated as meaningful when the original reference dataset was constructed in Dooley Jr. et al., 2019. In my first review, I described in detail the difficulty and problematic nature of making taxonomic determinations for specimens that occupy morphological and geologic spaces represented by both species. I also raised concerns related to the difference between how geographic association was treated in Dooley Jr. et al., 2019, (the source of the reference dataset used here) and how the present authors are treating geographic association. Although they pulled back on a few of their original taxonomic interpretations (I think as a consequence of the explicit criteria they developed for making such interpretations), the underlying problem remains. The authors did not respond direction to this concern. If they disagree with my evaluation, that is not clear, and I am not aware if they made any effort to defend their approach.

In my first review, I made the case for withholding judgement on specimens from Alberta. The authors chose to keep their identification of RAM P.97.7.1 as M. pacificus but did not provide explanation for their decision to reject my concern. It is possible that they felt the improved explanation of their method for reconstructing L:W of that specimen adequately addressed concerns. However, it is still the case that the proposed presence of M. pacificus in Alberta is based on evaluation of a single tooth with reconstructed dimensions, from a locality that has not previously produced M. pacificus remains, and with no clear chronological information. Among other things, the conservative approach to specimen identifications that the authors adopted in this revised manuscript seems inconsistent with this attribution. I am satisfied that the authors produced a reasonable reconstruction, but I don’t think it should be given the same weight as measurements from a complete tooth. In addition, if it can at least be attributed to either Irvingtonian or Rancholabrean (the deposits to which Dooley et al., 2019 restricted attributions of M. pacificus), then that should be clarified. Otherwise, the rationale for considering it a possible/likely occurrence of M. pacificus rather than one of the other named Mammut species that predate the Irvingtonian, some of which have narrow teeth like M. pacificus according to Dooley et al., 2019, is unclear.

In my initial review, I also recommended that the authors stop short of extending the range of M. pacificus into WA based on questionable attributions for two specimens. In this case, the revision dropped the assertions of M. pacificus in WA, likely based on application of their revised criteria for assignation to the taxon. However, in retaining the attribution to M. americanum for UWBM 88099 (rather than just attributing all specimens in WA to M. sp.) they created a new complication—a specimen with no mandibular tusks and m3 L:W “just outside of the 2[s.d.] for M. pacificus” (line 203) is now being recognized as the only M. americanum specimen (based on morphological data in their dataset at least) west of the Rocky Mountains. As with the Alberta specimen addressed above, if there is chronological information linking this specimen to either the Irvingtonian or Rancholabrean, it is not documented by the authors.

In both of these cases, which the authors present as the first morphological evidences of overlapping ranges of M. americanum and M. pacificus, the assignments are on fairly shaky ground. In both cases, it would set a better precedent to apply the argument presented by Dooley Jr. et al., 2019, for calling eastern specimens M. americanum even when their measurements are more consistent with M. pacificus:

“However, [narrow molars from outside of CA and ID] are infrequent and do not exhibit any clustering; each occurrence is found in an area in which the narrow tooth is an isolated outlier in a population more typical of M. americanum.” (Dooley et al., 2019).

Currently, UWBM 88099 is an isolated outlier in a large region otherwise interpreted as being inhabited in the late Pleistocene only by M. pacificus… and it is by no means conclusively M. americanum. I am not suggesting that it should be called M. pacificus here (although I am somewhat surprised that it wasn’t included with the CA specimens in the original dataset based on geographic proximity), but that it represents an unusually broad-toothed M. pacificus should be treated as perhaps more likely than it being the only identified M. americanum from the Pacific Northwest, but just outside of 2 s.d. for previously identified M. pacificus. After all, it would not be the only specimen identified as M. pacificus that currently falls outside of 2 s.d. and is very close in L:W to the other one.

Meanwhile, RAM P.97.7.1 is from a region where the only other specimen they evaluate pretty clearly does not fit with the description of M. pacificus. For the reasons listed above, even if the estimated reconstructed measurement is accurate, it would still be seemingly more parsimonious to treat this specimen as an outlier than as the northernmost occurrence of M. pacificus in a region known to have M. americanum. I assume that the authors do consider geography to be an important consideration for calling specimens M. pacificus. That is, I assume that they intend to follow the precedent set by Dooley Jr. et al., 2019, to treat the eastern specimens with narrow teeth (e.g., those from MO, IN, OH, NC, and FL) as representing some of the natural variation within M. americanum, variation that overlaps with the morphological variation in M. pacificus. I think considering the possibility that it represents a range extension of M. pacificus without formally “assigning” it to that taxon and without interpreting it as evidence of meaningful range overlap for the two Mammut taxa would be fully consistent with the approach for considering more eastern specimens and any others that are geographically dissociated with M. pacificus, so that is what I recommend here.

Validity of the findings

The inclusion of specimens outside of CA and ID is not itself problematic—that would in fact be expected for a long-lived species (hundreds of thousands of years at least) that included both states in its normal range—but with a taxonomy that implies a degree of morphological overlap and is therefore to some extent reliant on geographic association, sorting out the identification of any individual specimen with ambiguous geographic affinities requires that the morphological data be independently definitive. This is especially true if the new identifications are used to make assertions about niche partitioning and/or fluctuating ranges in intermediate geographical areas where both species have supposedly occurred. This is a comment I made previously, and the authors subsequently pulled back a little on their assertions and presented a more conservative (and more clearly justified) interpretation of the newly considered and re-considered specimens. The validity of range extensions for M. pacificus rests on the evidence overwhelming the recognition of significant morphological variation in M. americanum. Considering this, I don’t think the case for extending the range to Alberta is supported.

Meanwhile, extending the range into Mexico does not seem problematic and could even be biogeographically consistent with the topography of the American Cordillera (perhaps creating a geographic separation between the two Mammut taxa). However, the interpretation of M. pacificus in Alberta and the assertion of both Mammut taxa in the Pacific Northwest both seem likely to muddle rather than clarify the biogeographic picture for late-Pleistocene Mammut. In both cases, more data are needed in order to make the case for range overlap.

Referring to the Alberta specimens, the authors state, “Although our ability to relate the specimens in time is somewhat challenged, that does not diminish the significance of the observation of both taxa in the same geographic region” and then use this identification of M. pacificus in Alberta to support the case for “further complexity in movement of taxa through the interior of northern North America…” I think the data warrant a more conservative approach to interpreting the significance of this proposed instance of M. pacificus to avoid a cascade effect with subsequent authors re-interpreting specimens to expand the M. pacificus range almost indefinitely. After all, there are teeth within 1 s.d. of M. pacificus from MO, IN, OH, NC, and FL that are all currently in the reference dataset for M. americanum. It seems reasonable to treat RAM P97.7.1 as a possible occurrence of M. pacificus in Alberta, but nothing more than that.

Additional comments

I do not think the authors of this manuscript bear the burden here of justifying the recognition of the two mastodon species with significant morphological overlap (and possibly geographic overlap). However, in asserting a broader geographic range than originally proposed for M. pacificus, it is incumbent on the authors to recognize that the dataset they are using implicitly assumed that the range of M. pacificus was geographically restricted and exclusive. That is, specimens in California that fall within the range of those outside of CA and ID (considered M. americanum) were nevertheless considered M. pacificus due to their presence in that state and were included in the reference dataset. Likewise, specimens outside of CA and ID (considered there to be M. americanum) with measurements that fall within the range of those within those states were nevertheless considered M. americanum and were included in that reference dataset. The proposal that individual specimens outside of CA and ID are M. pacificus using the ranges, means, and standard deviations from that original datasets and thus extend the range of M. pacificus contradicts one of the assumptions on which the original dataset was constructed. Thus, it seems that range extensions of M. pacificus, especially into areas considered to be occupied by M. americanum should only be based on the presence of multiple specimens with unambiguous and conclusive morphological data.

Reviewer 3 ·

Basic reporting

No comment

Experimental design

I'm glad that the criteria for the separation of the two species has been clarified, the methods described, and some level of statistical testing used to identify specimens based on molar proportions.

Validity of the findings

The authors have not taken up my suggestion of an allometric analysis, given that molars and limb bones are both smaller and of different proportions in pacificus compared to americanum. I would recommend this for future work.

---

## Round 0.3 · accepted · Accept

Congratulations all! Thanks again for this nice contribution. All my best, Steven